# Diet and Disease Activity in Patients with Axial Spondyloarthritis: SpondyloArthritis and NUTrition Study (SANUT)

**DOI:** 10.3390/nu14224730

**Published:** 2022-11-09

**Authors:** Pascale Vergne-Salle, Laurence Salle, Anne Catherine Fressinaud-Marie, Adeline Descamps-Deplas, François Montestruc, Christine Bonnet, Philippe Bertin

**Affiliations:** 1Rheumatology Department, University Hospital of Limoges and Laboratory PEIRENE UR 22722 Institut OmegaHealth, 87042 Limoges, France; 2Endocrinology Department, University Hospital of Limoges and Inserm U1094, EpiMaCT—Epidemiology of Chronic Diseases in Tropical Zone, 87042 Limoges, France; 3eXYSTAT, 92240 Malakoff, France

**Keywords:** spondyloarthritis, diet, omega-3 polyunsaturated fatty acids, fiber, ultra-transformed food

## Abstract

Axial Spondyloarthritis (axSpA) patients with inflamed intestines have higher SpA activity. Diets that modulate microbiota may influence inflammation and SpA activity. Today, data concerning the impact of diet on SpA activity are scarce. SANUT was a single-center, noninterventional, cohort study that assessed dietetic profiles associated with SpA activity in axSpA. Demographic, clinical, SpA-related, quality of life (QoL), fatigue, physical activity, and dietary data were collected. SpA activity was assessed by Ankylosing Spondylitis Disease Activity Score (ASDAS) and by Bath Ankylosing Spondylitis Disease Activity Index (BASDAI). We assessed whether high SpA activity was associated with nutriment consumption. Between 12 February 2018 and 12 February 2020, 278 patients participated. High SpA activity, as measured by ASDAS and BASDAI, was significantly associated with higher body mass index and waist circumference, negative HLA-B27, lower QoL, higher fatigue, and higher digestive-symptom scores. Furthermore, high SpA activity, as measured by BASDAI, was associated with female sex, smoking status, patients who were not actively employed, reduced physical activity, and high intake of ultra-transformed foods, while high SpA activity, as measured by ASDAS, was associated with low intake of omega-3 PUFAs and fiber. Therefore, low intakes of omega-3 PUFAs and fiber, and high intake of ultra-transformed foods, are associated with high SpA activity.

## 1. Introduction

Axial spondyloarthritis (axSpA) is a chronic inflammatory disease that predominantly affects the axial skeleton and can also include peripheral arthritic and enthesitic involvement [1]. Extra-articular manifestations include anterior uveitis, psoriasis, and inflammatory bowel disease (IBD) [1].

About 50–60% of patients with ax SpA have subclinical inflammation of the intestinal mucosal barrier [2]. Increased intes tinal inflammation correlates with more active SpA disease [3,4]. Inflammation has also been linked to intestinal barrier disruption, allowing pathogenic antigens to translocate, inducing inappropriate and uncontrolled immune response [5]. This physiopathological mechanism has been proposed for chronic IBD [6]. In the intestines, the microbiota protects the mucosal barrier from pathogen invasion, metabolizes food into useful nutrients, and contributes to immune homeostasis [7].

SpA studies provide evidence of dysbiosis and abnormal intestinal microbiota; however, the abnormalities observed vary substantially with studies [8,9]. In animal models, microbiota modulate the innate immune system in several pathologies with various immune cells, including type 17 lymphocyte T cells (Th17) [7,10,11]. In SpA, Th17 has a central role in bowel inflammation and in SpA pathogenesis [12].

Several factors influence microbiota quality, thus modulating the innate immune system in the intestinal barrier, the most important being diet [13]. Among dietary constituents, short-chain fatty acids (SCFAs) from the fermentation of dietary fiber by commensal bacteria are essential for maintaining a healthy microbiota and digestive tract [14]. SCFAs play a major role in gut homeostasis through tissue repair, intestinal epithelial cell mucous secretion, and immunomodulatory properties promoting lymphocyte T regulators (Tregs) responses [14,15,16].

The concept of prebiotics, elaborated in 1995 and recently reviewed, is based on the premise that certain food constituents (fructo-oligosaccharides or polysaccharides) fermented by specific bacteria can, among other roles, reinforce the intestinal barrier and decrease the translocation (endotoxemia) of proinflammatory bacterial constituents and metabolites, such as lipopolysaccharides (LPS), peptidoglycans, and bacterial DNA [17,18].

Omega-3 polyunsaturated fatty acid (PUFAs) consumption is beneficial for treating/preventing chronic diseases, including rheumatoid arthritis and IBD, mainly because of their anti-inflammatory properties [19,20]. Indeed, PUFAs are precursors for resolvins, the lipidic mediators involved in resolving inflammation.

Currently, there is a paucity of data concerning the impact of diet on SpA activity. The objective of this study was to identify dietary constituents (mainly omega-3 PUFAs, fiber, refined sugar, ultra-transformed foods, and vitamin C) associated with SpA activity using qualitative and semi-quantitative food surveys.

## 2. Materials and Methods

### 2.1. Study Design

The SANUT study was a single-center, noninterventional, cohort study. The French ethics committee “CPP Ouest VI” (ID-RCB: 2017-A02044-49) approved the study.

### 2.2. Participants

Between 12 February 2018 and 12 February 2020, all patients who presented at the Rheumatology Department of the “Centre Hospitalier Universitaire (CHU)” at Limoges and those connecting online to the French SpA Association, “l’Association France Spondyloarthrite (AFS)”, were asked to participate. Patients older than 18 years and meeting the assessment criteria of the SpondyloArthritis International Society (ASAS) for axSpA were eligible [21]. Patients with IBD or a history of bariatric surgery, and those unable to complete the questionnaire, or those under tutorship or curatorship, were not eligible.

### 2.3. Data Collection

Data from eligible patients were collected during a consultation or online. Patient characteristics, nutritional and disease-related data, as well as quality of life (QoL), fatigue, and physical activity data were collected. Patient characteristics included demographic data (age, sex, and sociodemographic data), physical data (body mass index (BMI) and waist circumference), and smoking status. Disease-related data comprised a history and phenotype of SpA (radiographic or nonradiographic axial SpA), current treatments, human leukocyte antigen (HLA) B27 status, last C-reactive protein (CRP) level, and disease activity as assessed using the Bath Ankylosing Spondylitis Disease Activity Index (BASDAI) [22] and the Ankylosing Spondylitis Disease Activity Score (ASDAS) [23].

Nutritional data included typical food intake collected using a 60-item survey, the use of specific diets (such as diabetic, gluten-free, and reduced-fat diets), vitamin D intake, and nutritional supplements.

The digestive symptom questionnaire had six questions, one for each symptom: constipation/hard stools, diarrhea/frequent stools, stomach pains, gastroesophageal reflux/pyrosis, abdominal pain, and bloating. Each question was scored between 0 (never occurred) to 4 (always), with a maximum score of 24.

QoL data were collected using the disease-specific Ankylosing Spondylitis Quality of Life (ASQoL) instrument [24]. The ASQoL comprises 18 items with a maximum score of 18; higher scores indicate a diminished QoL. Fatigue was evaluated using the Functional Assessment of Chronic Illness Therapy-Fatigue (FACIT-F) instrument [25]. The FACIT-F instrument has 13 items with a maximum score of 52; lower scores indicate more fatigue.

Physical activity levels were assessed by the short version of the International Physical Activity Questionnaire (IPAQ; French version, July 2003), which measures physical activity in metabolic equivalents (MET) [26]. Patients were classified into low (<600 MET/min), moderate (≥600 but <1500 MET/min), and high activity (≥1500 MET/min) groups.

### 2.4. Food-Intake Survey and Calculation of the Indices for Fiber, Refined Sugars, Omega-3 PUFAs, Ultra-Processed Foods, and Vitamin C Consumption

Food-intake data were collected with a 60-item survey in 8 categories: meats and eggs (10 items); oils, butter, and dairy (14 items); fruits and vegetables (7 items); complex and refined carbohydrates (11 items); ultra-processed foods (3 items); drinks (8 items); spices (3 items); and sweets and chocolate (4 items). The survey was based on recommendations from the French National Federation for Regional Health Observatories (FNORS) and reported food-frequency questionnaires [27,28]. Habitual food consumption was evaluated daily, weekly, or monthly depending on the food type. The food-intake data were used to calculate nutritional indices for vitamin C (range 0–2.2), fiber (range 0–26), omega-3 PUFAs (range 0–6.5), refined sugars (range 0–15), and ultra-processed food consumption (range 0–13). These nutritional indices were calculated using the French food composition table (CIQUAL 2016 [29]). For refined sugars, we calculated the ratios of carbohydrates to fiber, starch to fiber, and sugar to fiber so as to differentiate highly and less-refined foods. The omega-3 PUFAs index was based on eicosapentaenoic acid and docosahexaenoic acid in foods. An omega-3 PUFA index score alone did not completely reflect its benefit. Therefore, an omega-6 score was calculated to modulate the omega-3 index. Ultra-processed foods were identified by the degree of transformation and the presence of colorants, flavor enhancers, and texturing agents.

### 2.5. Study Objectives and Outcomes

We hypothesized that the intake of certain foods would be associated with axSpA activity.

Our primary objective was to identify associations between the consumption of nutrients (vitamin C, fiber, omega-3 PUFAs, refined sugars, and ultra-processed foods) and SpA activity.

SpA activity was assessed using BASDAI and ASDAS. The overall BASDAI scores ranged from 0 to 10, with a BASDAI ≥ 4 indicating active SpA. For the analyses, patients were classified as either BASDAI ≥ 4 or <4. For ASDAS, patients were classified as either ASDAS < 2.1 (in remission/inactive or having moderately active SpA) or ASDAS ≥ 2.1 (having highly or extremely active SpA).

Our secondary objectives assessed whether consuming nutrients (vitamin C, fiber, omega-3 PUFAs, refined sugars, and/or ultra-processed food) were associated with QoL.

### 2.6. Statistical Analysis

Data are presented using descriptive statistics. Continuous variables are reported as means with standard deviations (SD) and/or medians with ranges. Categorical variables are reported as numbers with frequencies (%). Missing data were not replaced. Percentages were calculated based on available data.

The association between SpA activity, according to the BASDAI (index ≥ 4) and the ASDAS (score ≥ 2.1), were analyzed by univariate analysis. Variables found to be significantly associated with SpA activity, with a *p* < 0.05 (Chi^2^ test), were then analyzed in multivariate analyses. A backward selection method was used to select the variables. Variables with significance levels below 5% (Wald Chi^2^ test) were maintained in the model. The multivariate models were adjusted for gender and BMI. Odds ratios (ORs) were reported with 95% two-sided confidence intervals (CIs).

A nutritional score, using the consumption indices significantly associated with SpA activity using BASDAI ≥ 4 to define the score and ASDAS ≥ 2.1 to validate the score in the multivariate models, was constructed using maximum likelihood estimates from logistic regression. This score was included in the final multivariate model to model the probability of SpA activity.

All analyses were performed using SAS version 9.4 software (SAS Institute, Cary, NC, USA).

## 3. Results

### 3.1. Patient Characteristics

Between 12 February 2018 and 12 February 2020, 278 patients completed the study-specific survey: 211 from hospital recruitment and 67 online. The study population (*n* = 278) was mostly female (57.6%) with a mean age of 51.7 years (SD 12.6). The sociodemographic data, physical data, type of axSpA, HLA-B27 positivity, CRP, QoL, fatigue, digestive symptoms, smoking status, and physical activity are summarized in Table 1. Vitamin D intake, specific diets, use of nutritional supplements, and nutriment consumption indices are summarized in Table 2.

Of the 278 patients enrolled, 274 (98.5%) had data to determine the BASDAI score. BASDAI scores were ≥4 (active SpA) in 179 patients (65.3%) (Table 1). A total of 235 patients (84.5%) had the required data to determine the ASDAS. The ASDAS was ≥2.1 (highly active and extremely active) in 179 patients (76.2%) (Table 1).

### 3.2. Identification of Variables Associated with SpA Activity in Univariate Analysis

Univariate analyses identified variables associated with SpA activity, estimated by ASDAS ≥ 2.1 and BASDAI ≥ 4. 

In the univariate analysis, using ASDAS to assess SpA activity, higher values of BMI, increased waist circumference, negative HLA-B27, lower QoL (a higher ASQoL score) and more fatigue (a lower FACIT-F score) were significantly associated with increased SpA activity (ASDAS ≥ 2.1). Lower omega-3 PUFAs and lower fiber intake were significantly associated with increased SpA activity. The calculated nutritional score was associated with increased SpA activity with an odds ratio (OR) of 2.72. A higher digestive symptom score and more epigastralgia, gastroesophageal reflux, abdominal pain, and bloating were also associated with higher SpA activity (Figure 1).

Similarly, in the univariate analysis, using BASDAI to assess SpA activity, female sex, smoking status, a lack of active employment, high values of BMI (≥30 kg/m^2^), increased waist circumference, negative HLA-B27, low levels of physical activity, lower QoL (a higher ASQoL score) and more fatigue (a lower FACIT-F score) were significantly associated with increased SpA activity (BASDAI ≥ 4). Higher ultra-transformed food intake was significantly associated with increased SpA activity. Lower omega-3 PUFA intake was not significantly associated with SpA activity, but results were close to a level of significance. As for the ASDAS, the calculated nutritional score was associated with increased SpA activity as measured by BASDAI, with an OR of 2.72. Higher digestive symptom scores and more epigastralgia, gastroesophageal reflux, abdominal pain, and bloating were also associated with higher SpA activity (Figure 2).

Comorbidities, such as diabetes and hypercholesterolemia, were not associated with disease activity in univariate analysis, but the number of patients with diabetes and hypercholesterolemia was low (*n* = 11 and 33, respectively).

### 3.3. Identification of Variables Associated with SpA Activity in Multivariate Analysis

The multivariate analysis showed that higher SpA activity with ASDAS ≥ 2.1, was significantly associated with higher BMI and negative HLA-B27 (Table 3). 

When high SpA activity was defined as BASDAI ≥ 4, multivariate analysis found a higher digestive symptom score, a lack of active employment, and negative HLA-B27 (Table 3) to be significantly associated. Higher SpA activity (BASDAI ≥ 4) was significantly associated with higher consumption of ultra-transformed foods.

### 3.4. Analysis of the Nutritional Factors Associated with Quality of Life

In the univariate analysis, a lower QoL (a score ≥ 8, the median value) was significantly associated with the female gender (OR 0.55 (95% CI 0.32, 0.93) *p* = 0.03); tobacco use (OR 0.38 (95% CI 0.19, 0.78) *p* = 0.02); a higher BMI (OR 0.15 (95% CI 0.06, 0.38) *p* = 0.004); HLA-B27 positivity (OR 2.9 (95% CI 1.6, 5.4) *p* = 0.0007); and a higher overall digestive symptom score (OR 0.86 (95% CI 0.81, 0.92)). An inverse association between QoL and omega-3 PUFA consumption was observed: the lower the QoL, the lower the consumption of omega-3 PUFAs (OR 1.4 (95% CI 1.04, 1.8), *p* = 0.02). The multivariate analysis confirmed that a lower QoL was associated with a higher BMI (*p* = 0.005), HLA-B27 positivity (*p* = 0.003), and a higher overall digestive symptom score (*p* = 0.0002).

### 3.5. Post-Hoc Analyses to Define a Nutritional Score

The univariate analysis of higher SpA activity showed that omega-3 PUFAs, fiber, and ultra-transformed foods were statistically significant. A nutritional score combining these consumption indices was built using logistic regression estimations as: Nutritional score =1.8165 − 0.2428 ∗ Omega_3 − 0.1044 ∗ Fiber + 0.0764 ∗ UT. This score was tested in the same multivariate models (Table 4). The nutritional score was significant in both models.

The multivariate analysis showed that higher SpA activity with ASDAS ≥ 2.1, was significantly associated with a higher digestive symptom score (*p* = 0.03), a lack of active employment (*p* = 0.03), and negative HLA-B27 (*p* = 0.03) (Table 4). Higher SpA activity (ASDAS ≥ 2.1) was significantly associated with nutritional score (OR 3.1 (95% CI 1.4, 6.8), *p* = 0.006).

When high SpA activity was defined as BASDAI ≥ 4, multivariate analysis found a higher digestive symptom score (*p* = 0.007), a lack of active employment (*p* = 0.008), and negative HLA-B27 (*p* = 0.02) (Table 4) to be significantly associated. Higher SpA activity (BASDAI ≥ 4) was significantly associated with nutritional score (OR 3.1 (95% CI 1.5, 6.6), *p* = 0.003).

Using these exploratory multivariate models, we estimated the probability of high SpA activity in the worst- and best-case scenarios.

When the patient is HLA-B27 positive, actively employed, and has a low digestive symptom score of 0 and a low nutritional score of −0.84, then the probability of high SpA activity is 7% for ASDAS and 3% for BASDAI. Conversely, when the patient is HLA-B27 negative, not actively employed, and has a high digestive symptom score of 17 and a high nutritional score of 1.64, then the probability of a high SpA activity is 97% for ASDAS and for BASDAI.

## 4. Discussion

Among nutriments, lower fiber and omega-3 PUFA consumption were associated with increased SpA activity as measured by ASDAS in univariate analysis, while consuming ultra-processed foods was associated with high SpA activity by BASDAI. Only the association between ultra-processed food consumption and BASDAI score persists in multivariate analysis.

Few studies have assessed the dietary impact on SpA activity. Sundstrom et al. evaluated the impact of diet on SpA activity and gastrointestinal symptoms [30]. The study found no association between diet and SpA activity. However, the study only assessed SpA activity by BASDAI and analyzed 111 patients: 77% males, and 100% with axial radiographic SpA. This study population is dissimilar to our population because we also included nonradiographic axSpA. Our study found that patients with gastrointestinal pain had increased SpA activity. Similarly, in our study, SpA activity was significantly associated with epigastralgia, gastroesophageal reflux, abdominal pain, and bloating. A case control study assessed the effect of food intake and air pollution exposure on SpA activity using BASDAI [31]. SpA patients, compared to healthy subjects, consumed significantly more energy, carbohydrates, fats, and vitamins. Associations between nutriments consumed and SpA activity were not reported.

We focused on omega-3 PUFAs as a precursor to resolvins implicated in the resolution of inflammation [19,20], and we found that omega-3 consumption was associated with SpA activity. There is growing evidence supporting the benefit of consuming omega-3 PUFAs in several chronic diseases [19,32,33,34]. Omega-3 PUFAs modulate gut microbiota, help maintain the intestinal wall, interact with the immune system, and ultimately reduce inflammation. Gut dysbiosis can cause and exacerbate metabolic endotoxemia by increasing blood levels of LPS [35]. In mice, increased consumption of omega-3 PUFAs reduces LPS blood levels and inflammation [35]. The analysis of the gut microbiota in these mice found fewer LPS-producing/proinflammatory bacteria (such as *Proteobacteria*) and more LPS-suppressing/anti-inflammatory bacteria (such as *Bifidobacterium*). Supplementing omega-3 PUFAs stimulates the growth of bacteria that produce anti-inflammatory SCFAs [36]. Butyrates are vital for maintaining the intestinal barrier by regulating the assembly of tight joints [14,15,16]. Low SCFA levels are associated with several inflammatory diseases, including IBD [37,38]. Omega-3 PUFAs, as immune modulates, induce the production of anti-inflammatory factors, such as IL-10, promote Treg differentiation, and regulate Th17 cells, which secrete the proinflammatory cytokine, interleukin 17 (IL-17) [36]. In the gut, increased Treg differentiation and lower IL-17 levels may reduce inflammation [36].

Fiber consumption benefits patients with various inflammatory diseases, including IBD [39], rheumatoid arthritis [40], and axSpA (in our study). We included various types of fiber in the food-intake survey since not all types of fiber have prebiotic activity [41]. In our study, patients needed to consume various types of fiber to obtain a high score. Favorable microbiota modulation is linked to consuming diverse types of dietary fiber. Indeed, most bacteria feed on a multitude of prebiotics. Cross-feeding between bacteria exists [41]. Overall, the variety of bacterial metabolites depends on the quality of the microbiota. Identifying specific types of dietary fiber related to inflammation without metabolomic studies is difficult. We found that a low consumption of fiber was associated with higher SpA activity. Diminished bacterial diversity by insufficient fiber consumption, with altered prebiotic activity, causes dysbiosis. Dysbiosis promotes LPS production and disrupts the intestinal barrier that clinically manifests by an upsurge in the expression of chronic inflammatory diseases [42].

Recently, a monocentric observational study assessed the Mediterranean diet in 161 patients with axial SpA [43]: 81 received nutritional advice for the Mediterranean diet. After 6 months, both SpA activity (by ASDAS-CRP) and adherence to the diet had significantly improved in the patients with nutritional advice. Despite the large number of patients lost to follow-up, these results are in line with our results since the Mediterranean diet increases omega-3 PUFA and fiber consumption.

Currently, there is growing concern about the increased consumption of ultra-processed foods [44,45,46]. Industrially, whole foods are refined and fractionated into their components and then recombined, producing ultra-processed foods. These foods with poor nutritional density lack bioactive components and protective fiber. To improve the sensorial aspects of ultra-processed foods and increase shelf-life, additives are required. We found that increased consumption of ultra-processed food was associated with increased SpA activity by BASDAI, both in univariate and multivariate analysis. Food additives, including fractionated sugars, emulsifiers, salt, gluten, and microbial transglutaminase disrupt the “mucosal firewall” [47]. Indeed, the increased incidence of autoimmune diseases has been accompanied by an increase in the use and consumption of additives [47]. For example, the recent increase in celiac disease incidence may partly be due to the increased intake of microbial transglutaminase, an autoantigen for celiac disease [47,48]. Microbial transglutaminase increases intestinal permeability, deteriorating the “mucosal firewall”, and stimulating luminal bacteria proliferation [48]. Thus, there is scientific rational for our observation that consuming more ultra-processed foods was associated with higher SpA activity.

Moreover, vitamin C intake did not correlate with SpA activity in our study, nor did refined sugar. However, vitamin C intake was negatively associated with fatigue (by FACIT-F), an important component of SpA. However, vitamin C intake, using a food-intake questionnaire, does not reflect the amount available. Indeed, vitamin C is very water-soluble, with its excess rapidly cleared by the kidneys, making it difficult to correlate intake with inflammation [49].

In our study, SpA activity was associated with BMI, as well as with digestive symptoms, as previously reported [30,50]. Poor QoL was associated with increased BMI and digestive symptoms and with lower omega-3 PUFA consumption.

To our knowledge, this is the first time that nutriment consumption has been associated with SpA activity. A major strength of our study is that the studied population included radiographic and nonradiographic SpA. We assess SpA activity using the BASDAI index, but also with the ASDS-CRP, including a biologic marker of inflammation. Dietary habits were collected using a qualitative and semi-qualitative food-intake questionnaire. The nutritional indices were calculated using the French food composition table (CIQUAL 2016 [29]), balancing the ratio of omega-6 to omega-3, requiring the intake of diverse types of fiber to obtain a high score, and identifying ultra-processed foods. However, our study, as with other studies based on food surveys, is limited by the declarative data from patients with inherent recall bias and by missing data. Our study did not analyze for calprotectin, nor did we study microbiota. We do not have an explanation for the discrepancy between ASDAS and BASDAI associations with nutriment consumption. These two validated disease activity scores show a good correlation, but they are different. The ASDAS score includes a blood marker of inflammation and may better reflect the inflammatory process, while the BASDAI score may be more affected by patient-reported outcomes. However, the association between higher BASDAI score and lower omega-3 PUFA consumption was close to significance.

Omega-3 PUFAs and fiber are only associated with SpA activity in the univariate analysis. It is possible that our study lacks power or that the adjustment according to BMI interferes with the association. Indeed, it is probable that patients with a high BMI consume fewer omega-3 PUFAs and less fiber [51]. In contrast, the calculated nutritional score that includes omega-3 PUFAs, fiber, and ultra-transformed foods is associated with SpA activity in multivariate analysis. This nutritional score provides a better global nutritional profile of patients. 

Finally, a larger, multicenter study assessing the predictive value of our nutritional score on SpA activity would be interesting. Furthermore, a large interventional, multicentric study assessing an anti-inflammatory diet, rich in omega-3 PUFAs and fiber, and low in ultra-transformed foods, is required to confirm our results.

## 5. Conclusions

This study is the first to show the association between diet and SpA activity. Low intakes of omega-3 PUFAs and fiber were associated with higher SpA activity in univariate analysis, and high intakes of ultra-transformed foods were associated with increased SpA activity in multivariate analysis. The nutritional score combining the three nutriments was associated with SpA activity in multivariate analysis. These results are consistent with the role of omega-3 PUFAs and fiber in reducing immuno-inflammatory reactions and maintaining the intestinal barrier. In contrast, for ultra-transformed foods, the underlying mechanism requires further research.

## Figures and Tables

**Figure 1 nutrients-14-04730-f001:**
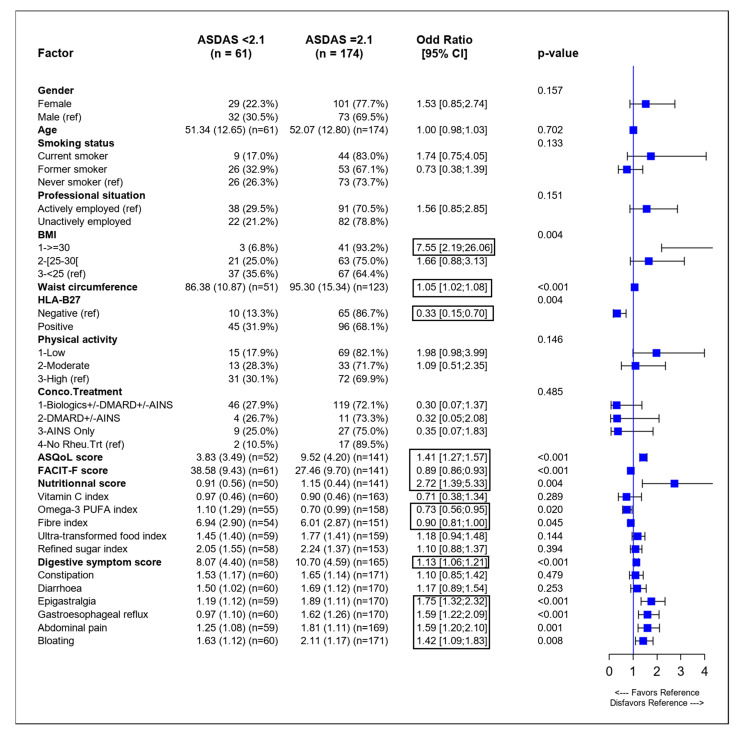
Forest plot of univariate analysis of variables associated with higher SpA activity evaluated using ASDAS. ASDAS: Ankylosing Spondylitis Disease Activity Score, ASQoL: Ankylosing Spondylitis Quality of Life, BASDAI: Bath Ankylosing Spondylitis Disease Activity Index, BMI: body mass index, CI: confidence intervals, HLA-B27: human leucocyte antigen B27, PUFA, polyunsaturated fatty acids, ref: reference value. Significant odds ratios are boxed.

**Figure 2 nutrients-14-04730-f002:**
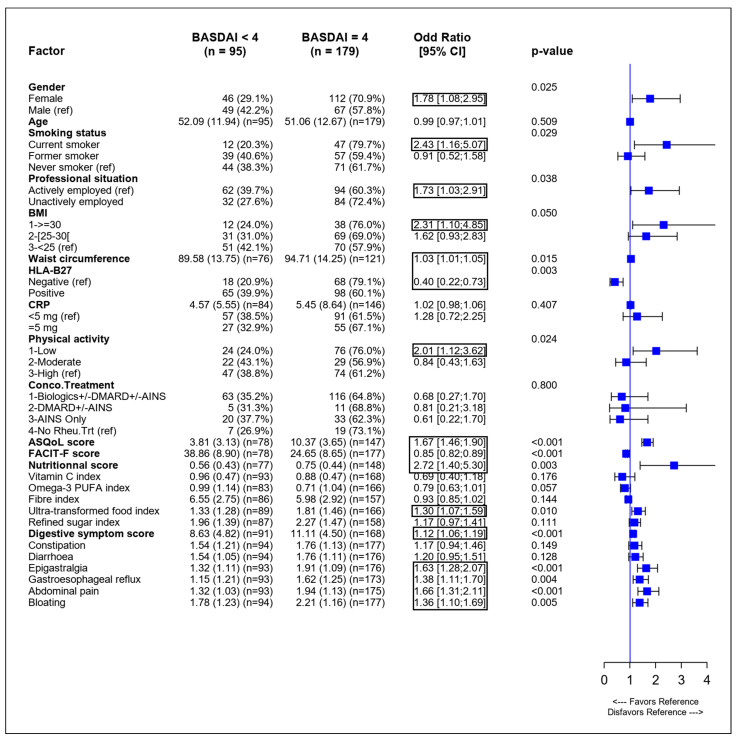
Forest plot of univariate analysis of variables associated with higher SpA activity evaluated using BASDAI. ASDAS: Ankylosing Spondylitis Disease Activity Score, ASQoL: Ankylosing Spondylitis Quality of Life, BASDAI: Bath Ankylosing Spondylitis Disease Activity Index, BMI: body mass index, CI: confidence intervals, CRP: C-reactive protein, DMARD: disease-modifying antirheumatic drugs, HLA-B27: human leucocyte antigen B27, NSAID: nonsteroidal anti-inflammatory drugs, PUFA, polyunsaturated fatty acids, ref: reference value. Significant odds ratios are boxed.

**Table 1 nutrients-14-04730-t001:** Baseline characteristics.

		Patients with Data, *n*	Details
Age (years), mean (SD)		278	51.7 (12.6)
Sex (female), *n* (%)		278	150 (57.6)
Height (cm), mean (SD)		277	167.4 (9.2)
Weight (kg), mean (SD)		275	74.1 (16.1)
BMI (kg/m^2^), mean (SD)		274	26.4 (5.3)
Waist circumference (cm), mean (SD)		200	92.9 (14.4)
Disease duration (years), mean (SD)		263	13.9 (10.6)
Radiographic axSpA, *n* (%)		278	132 (47.5)
Nonradiographic axSpA, *n* (%)		278	146 (52.5)
Positive for HLA-B27, *n* (%)		252	165 (65.5)
CRP (mg/L), mean (SD)		233	5.2 (7.6)
	CRP < 5 mg, *n* (%)		149 (63.9)
	CRP ≥ 5 mg, *n* (%)		84 (36.1)
ASDAS (using CRP), mean (SD)		235	2.6 (0.8)
	ASDAS < 2.1, *n* (%)		56 (23.8)
	ASDAS ≥ 2.1, *n* (%)		179 (76.2)
BASDAI score, mean (SD)		274	4.6 (1.9)
	BASDAI < 4, *n* (%)		95 (34.7)
	BASDAI ≥ 4, *n* (%)		179 (65.3)
ASQoL, mean (SD)		226	8.1 (4.7)
FACIT-F, mean (SD)		276	29.4 (10.9)
Digestive symptom score, mean (SD)		261	10.2 (4.8)
Professional activity, *n* (%)		276	
	Actively employed		157 (56.9)
	Not actively employed		119 (43.1)
Smoking status, *n* (%)		273	
	Former smoker		97 (35.5)
	Current smoker		59 (21.6)
	Never smoker		117 (42.9)
Treatments, *n* (%)		278	
	NSAIDs		129 (46.4)
	sDMARDs		30 (10.8)
	Anti-TNF		150 (54.2)
	Other biotherapies		33 (11.9)
	Steroids		16 (5.8)
	Anticholesterolemic		33 (11.9)
	Antidiabetic		11 (4.0)
IPAQ, mean (SD)		276	2789 (4113)
	Low physical activity, *n* (%)		102 (37.0)
	Moderate physical activity, *n* (%)		52 (18.8)
	High physical activity, *n* (%)		122 (44.2)

Anti-TNF: anti-tumor necrosis factor, ASDAS: Ankylosing Spondylitis Disease Activity Score, ASQoL: Ankylosing Spondylitis Quality of Life, BASDAI: Bath Ankylosing Spondylitis Disease Activity Index, BMI: body mass index, CRP: C-reactive protein, sDMARDs: synthetic disease-modifying antirheumatic drugs, FACIT-F: Functional Assessment of Chronic Illness Therapy-Fatigue, HLA-B27: human leucocyte antigen B27, IPAQ: International Physical Activity Questionnaire, NSAIDs: nonsteroidal anti-inflammatory drugs, SD: standard deviation, axSpA: axial spondyloarthritis.

**Table 2 nutrients-14-04730-t002:** Dietary data collected, and indices and nutritional scores calculated during the SANUT study.

		Patients with Data, *n*	Details
Vitamin D supplementation, *n* (%)		145	
	None		53 (36.6)
	Annually		28 (19.3)
	Every 3 months		29 (20.0)
	Monthly		13 (9.0)
	Twice monthly		9 (6.2)
	Daily		13 (9.0)
Specific diets, *n* (%)		275	
	None		214 (77.8)
	Diabetic diet		6 (2.2)
	Reduced-fat diet		12 (4.4)
	Gluten-free diet		18 (6.5)
	Other diets		25 (9.1)
Nutritional supplement intake, *n* (%)		275	52 (18.9)
Consumption indices			
Vitamin C index		264	
	Mean (SD)		0.9 (0.5)
	Median (IQR)		0.8 (0.6, 1.2)
Omega-3 PUFA index		251	
	Mean (SD)		0.8 (1.1)
	Median (IQR)		0.4 (0.0, 1.3)
Fiber index		243	
	Mean (SD)		6.2 (2.9)
	Median (IQR)		6.0 (4.0, 7.5)
Ultra-transformed foods index		257	
	Mean (SD)		1.6 (1.4)
	Median (IQR)		1.1 (0.6, 2.5)
Refined sugar index		246	
	Mean (SD)		2.2 (1.4)
	Median (IQR)		2.0 (1.0, 3.0)

IQR: interquartile range, PUFA: polyunsaturated fatty acids, SD: standard deviation.

**Table 3 nutrients-14-04730-t003:** Multivariate analysis of variables associated with higher SpA activity, with ASDAS ≥ 2.1 or BASDAI ≥ 4.

		ASDAS ≥ 2.1 (*n* = 214)		BASDAI ≥ 4 (*n* = 220)	
		OR (95% CI)	*p*	OR (95% CI)	*p*
Smoking status		Not significant in the multivariate model.		Not significant in the multivariate model.	
Sex		Not significant in the multivariate model.		Not significant in the multivariate model.	
BMI			0.005	Not significant in the multivariate model.	
	<25 (ref)				
	(25;30)	1.9 (0.9, 3.8)			
	≥30	7.1 (2.0, 25.0)			
Physical activity		Not significant in the multivariate model.		Not significant in the multivariate model	
Digestive symptom score		Not significant in the multivariate model.		1.14 (1.07, 1.23)	0.0001
Professional situation		Not significant in the multivariate model.			0.003
	Actively employed (ref)				
	Not actively employed			2.7 (1.4, 5.1)	
HLA-B27		0.004 0.3 (0.1, 0.7)			0.008
	Negative (ref)			0.4 (0.2, 0.8)	
	Positive				
Omega-3 PUFA index		Not significant in the multivariate model		Not significant in the multivariate model	
Fiber index		Not significant in the multivariate model		Not significant in the multivariate model	
Ultra-transformed foods index		Not significant in the multivariate model		1.4 (1.1, 1.7)	0.01

ASDAS: Ankylosing Spondylitis Disease Activity Score, BASDAI: Bath Ankylosing Spondylitis Disease Activity Index, BMI: body mass index, CI: confidence interval, HLA-B27: human leucocyte antigen B27, OR: odds ratio, PUFA: polyunsaturated fatty acid, ref: reference value.

**Table 4 nutrients-14-04730-t004:** Multivariate analysis of variables associated with higher SpA activity, with ASDAS ≥ 2.1 or BASDAI ≥ 4.

		ASDAS ≥ 2.1 (*n* = 168)		BASDAI ≥ 4 (*n* = 192)	
		OR (95% CI)	*p*	OR (95% CI)	*p*
Smoking Status		Not significant in the multivariate model.		Not significant in the multivariate model.	
Sex		Not significant in the multivariate model.		Not significant in the multivariate model.	
BMI		Not significant in the multivariate model.		Not significant in the multivariate model.	
Physical activity		Not significant in the multivariate model.		Not significant in the multivariate model	
Digestive symptom score		1.1 (1.01, 1.2)	0.03	1.1 (1.06, 1.2)	0.007
Professional situation		0.03			0.008
	Actively employed (ref)				
	Not actively employed	2.5 (1.1, 5.7)		3.7 (1.7, 7.8)	
HLA-B27		0.03 0.4 (0.2;0.9)			0.02
	Negative (ref)			0.4 (0.2, 0.9)	
	Positive				
Nutritional score *		3.1 (1.4, 6.8)	0.006	3.1 (1.5, 6.6)	0.003

ASDAS: Ankylosing Spondylitis Disease Activity Score, BASDAI: Bath Ankylosing Spondylitis Activity Index, BMI: body mass index, CI: confidence interval, HLA-B27: human leucocyte antigen B27, OR: odds ratio. * score = 1.8165 − 0.2428 ∗ Omega_3 − 0.1044 ∗ Fiber + 0.0764 ∗ UT.

## Data Availability

The data collected to support the results of this study are available from the corresponding author upon reasonable request.

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
