# Peer review of "Diet and Disease Activity in Patients with Axial Spondyloarthritis: SpondyloArthritis and NUTrition Study (SANUT)"

_nutrients, 2022, doi:10.3390/nu14224730_

Round 1
Reviewer 1 Report
This is an interesting article on a topic that is gaining importance in the clinical setting: the effect of diet on disease activity and on objective and subjective indexes of spondylarthritis. The article is well written (excepted introduction that is less clear) and the discussion is well argued. Also, a good exposition about different questionnaire used was built. It investigates the impact of dietary habits (included smoke, physical activity, and BMI) and microbiota involvement on different aspects of Spondylarthritis, considering disease perception. Many aspects (objective and subjective) of disease are analyzed and different interesting correlation with demographic data were found.
However, I have some comments:
- Do you have data about the correlation with radiological progression of the disease?
- Did you observe differences among different phenotype of axial involvement SpA (PsA, ReA, enteropathic SpA…)?
- Comorbidities, such as arterial hypertension, hypercholesterolemia or diabetes, and some metabolic and nutritional factors (serum lipid or glucose levels) were not discussed. Are these factors valuated?
- In Figure 1 and Figure 2, I would better underline significative data
- HLA-B27 negativity is associated with higher disease activity, measured both with ADSAS and BASDAI but HLA-B27 positivity is related to lower QoL; do you have explanation or hyptotesis about this discrepancy? Are diet and drugs involved?
-Were patients with gastrointestinal symptoms studied with endoscopy that demonstrated a subclinical inflammation?
Other comments:
Page 1 Line 36: I suggest “enthesite involvement” and “extra-articular manifestations”
Page 1 Line 38: I suggest “about 50-60% of patients with SpA presents…”
Page 1 Line 41: pathogenic or pathogenetic?
Page 2 Line 48-49: I suggest “In SpA, Th 17 has a central role in bowel inflammation and in SpA pathogenesis.”
Page 4 Line 162: patientpatients
Author Response
Response to reviewer 1 comments :
- Do you have data about the correlation with radiological progression of the disease?
We have data concerning the presence or absence of radiographic lesions (more precisely sacroiliac joint involvement), with two groups of radiographic and non radiographic spondyloarthtritis. But, we don’t have data concerning the radiological progression of the disease, because the velocity of disease progression and severity are at variance and often slow. Otherwise, the potential impact of earlier treatment and the best drugs or combinations of drugs for preventing radiographic progression in spondyloarthritis are yet to be determined. The clinical symptoms, including pain, stiffness, function and disease activity are considered even more important.
- Did you observe differences among different phenotype of axial involvement SpA (PsA, ReA, enteropathic SpA…)?
Patient with psoriatic arthritis were not included because this phenotype is considered as peripheral spondyloarthritis with a different clinical evaluation and the inclusion criteria were axial spondyloarthritis. Nor do we have included IBD (it is specified in participants section page 2 line 78) because it could introduce a bias to evaluate the impact of diet on disease activity.
- Comorbidities, such as arterial hypertension, hypercholesterolemia or diabetes, and some metabolic and nutritional factors (serum lipid or glucose levels) were not discussed. Are these factors valuated?
Comorbidities data were collected as medical history and specific treatments: anticholesterolemic and antidiabetic drugs (shown in table 1). We have also taken into consideration the specific diets : diabetic diet and reduced fat diet (shown in table 2). But, few patients were concerned: 33 with hypercholesterolemia and 11 with diabetes. No difference in disease activity was observed between patients with diabetes and patients without diabetes, nor between patients with hypercholesterolemia and patients without hypercholesterolemia. We analyzed only variables associated with spondyloarthritis activity.
A short paragraph is added in the results section : page 8 line 220-222.
- In Figure 1 and Figure 2, I would better underline significative data
I agree with the reviewer’s comment. But, we wanted to keep data of concern, even non significative, as gender, smoker status, physical activity, disease modifying anti-rheumatic drugs, etc…, data that could have influenced disease activity.
Significant odds ratio are boxed in Figure 1 and 2
- HLA-B27 negativity is associated with higher disease activity, measured both with ADSAS and BASDAI but HLA-B27 positivity is related to lower QoL; do you have explanation or hyptotesis about this discrepancy? Are diet and drugs involved?
HLA-B27 negativity is more frequent in radiographic spondyloarthritis phenotype due to the assessment of Spondyloarthritis International Society (ASAS) criteria for axSpA. This phenotype of spondyloarthritis is characterized by higher CRP and a higher disease activity.
The quality of life may be influenced by many factors as pain, fatigue, disability and emotional factors, ie patient-reported outcomes which are more altered in non radiographic spondyloarthritis, the phenotype with HLAB27 positivity.
-Were patients with gastrointestinal symptoms studied with endoscopy that demonstrated a subclinical inflammation?
The patient with gastrointestinal symptoms didn’t have an endoscopy at the time of the inclusion in the study. A small percentage of them had a history of negative endoscopy for IBD.
Other comments:
- Page 1 Line 36: I suggest “enthesite involvement” and “extra-articular manifestations”.
Corrections have been made in the text
- Page 1 Line 38: I suggest “about 50-60% of patients with SpA presents…”
Corrections have been made in the text
- Page 1 Line 41: pathogenic or pathogenetic?
I think It is pathogenic antigens
- Page 2 Line 48-49: I suggest “In SpA, Th 17 has a central role in bowel inflammation and in SpA pathogenesis.”
Corrections have been made in the text
- Page 4 Line 162: patientàpatients : I don’t understand
Reviewer 2 Report
The study confirms the knowledge that mediterranean diet is useful. A prospective trial will be needed to confirm the findings.
It is also a self fulfilling prophecy, healthy living people eat healthier and feel better,
It is unclear why only ASDAS is correlated with this food intake and not the BASDAI.
Author Response
Response to reviewer 2 comments
The study confirms the knowledge that mediterranean diet is useful. A prospective trial will be needed to confirm the findings.
It is also a self fulfilling prophecy, healthy living people eat healthier and feel better,
It is unclear why only ASDAS is correlated with this food intake and not the BASDAI.
We don’t have an explanation for the discrepancy between ASDAS and BASDAI associations with nutriment consumption. These two validated disease activity scores showed a good correlation but are different. ASDAS score includes a blood marker of inflammation and may better reflect the inflammatory process, while BASDAI may be more affected by patient-reported outcomes. However, the association between higher BASDAI score and lower omega-3 PUFA consumption was close to significance.
These hypothesis have been added in the text, in the “discussion” section: page 12 line 371-377.
